# Efficient Sorbitol Producing Process through Glucose Hydrogenation Catalyzed by Ru Supported Amino Poly (Styrene-*co*-Maleic) Polymer (ASMA) Encapsulated on γ-Al₂O₃

**Jing Zhao [1], Xiaorui Yang [1], Wei Wang [1] , Jinhua Liang [1], Yasin Orooji [2] , Chaowen Dai [1], Xiaomin Fu [1], Yunsong Yang [1], Wenlong Xu [1,\*] and Jianliang Zhu [1,\*]**

[1] College of Biotechnology and Pharmaceutical Engineering, Nanjing Tech University (Nanjing Tech), 30 South Puzhu Road, Nanjing 211800, China; zhaojing1112@163.com (J.Z.); yangxiaorui@njtech.edu.cn (X.Y.); Wangwei007@njtech.edu.cn (W.W.); jhliang@njtech.edu.cn (J.L.); daichaowen6284@163.com (C.D.); xmf0824@163.com (X.F.); yys19961028yys@163.com (Y.Y.)

[2] College of Materials Science and Engineering, Nanjing Forestry University, 159 Longpan Road, Nanjing 210037, China; yasin@njfu.edu.cn

\* Correspondence: constantinos@njtech.edu.cn (W.X.); jlzhu@njtech.edu.cn (J.Z.); Tel.: +86-187-6168-0873 (W.X.); +86-139-0516-2748 (J.Z.)

**Abstract:** In this work, a core-shell-like sphere ruthenium catalyst, named as 5%Ru/γ-Al₂O₃@ASMA, has been successfully synthesized through impregnating the ruthenium nanoparticles (NPs) on the surface of the amino poly (styrene-*co*-maleic) polymer (ASMA) encapsulating γ-Al₂O₃ pellet support. The interaction between the Ru cations and the electro-donating polymer shell rich in hydroxyl and amino groups through the coordination bond would guarantee that the Ru NPs can be highly dispersed and firmly embedded on the surface of the support. In addition, the solid sphere γ-Al₂O₃ pellet could serve as the core to support the resulted catalysts applied in the flow process in a trickle bed reactor to promote the productivity. The resulted catalyst 5%Ru/γ-Al₂O₃@ASMA can be applied efficiently in the glucose hydrogenation and presents a steadfast sorbitol yield of almost 90% both in batch reactor and the trickle bed reactor, indicating the potential feasibility of the core-shell-like catalyst in the efficient production of sorbitol.

**Keywords:** core-shell; organic-inorganic hybrid; ruthenium catalyst; hydrogenation; d-glucose; targeting effect

## 1. Introduction

With the depletion of the crude-oil reservoirs and fossil fuels worldwide, the demand for the sustainable chemical resources has verified its feasibility to exert the global energy reform and environmental protections [1,2]. As a high-value hexitol derived from the facile biomass glucose [3,4], ranked as one of the top twelve biobased building blocks, investigation on the production of sorbitol efficiently has become more desirable both industrially and academically. It is worth noting that nearly all of the current sorbitol preparation methods have been fixed on the glucose hydrogenation with different heterogenous catalysts [5]. Therefore, the selection of an efficient and environmentally friendly catalyst plays the critical role, especially in a highly efficient glucose hydrogenation procedure, for example, with the application of three phase trickle bed reactor.

Generally, the most typical catalyst applied industrially in this field should be dedicated to the Raney Ni catalyst [6,7], which is highly active and has a low price but suffers from the active sites leaching

and is potentially hazardous to the sanitation and hygiene. Therefore, the utilization of noble metal catalysts, such as Ru based catalyst, has drawn into people's attention [3,8–22]. However, bearing in mind that glucose is not stable and is easy to be isomerized with the effect of high temperatures [23]. In addition, the resulted sorbitol could further yield mannitol and even short chain polyhydric alcohols and alkanes due to the deep hydrogenolysis (see Scheme 1) [10,24–27]. In order to restrain the side reaction to the target production of sorbitol, the demand for the Ru catalyst, especially the suitable support, becomes the center for this investigation, because the catalytic performance of these catalysts in the hydrogenation of glucose to sorbitol depends on not only the dispersion of Ru active sites [3], but also on the stability for resisting the sites sintering and leaching as well as the facile textural properties for molecular flux in the catalyst [10,16,28–31].

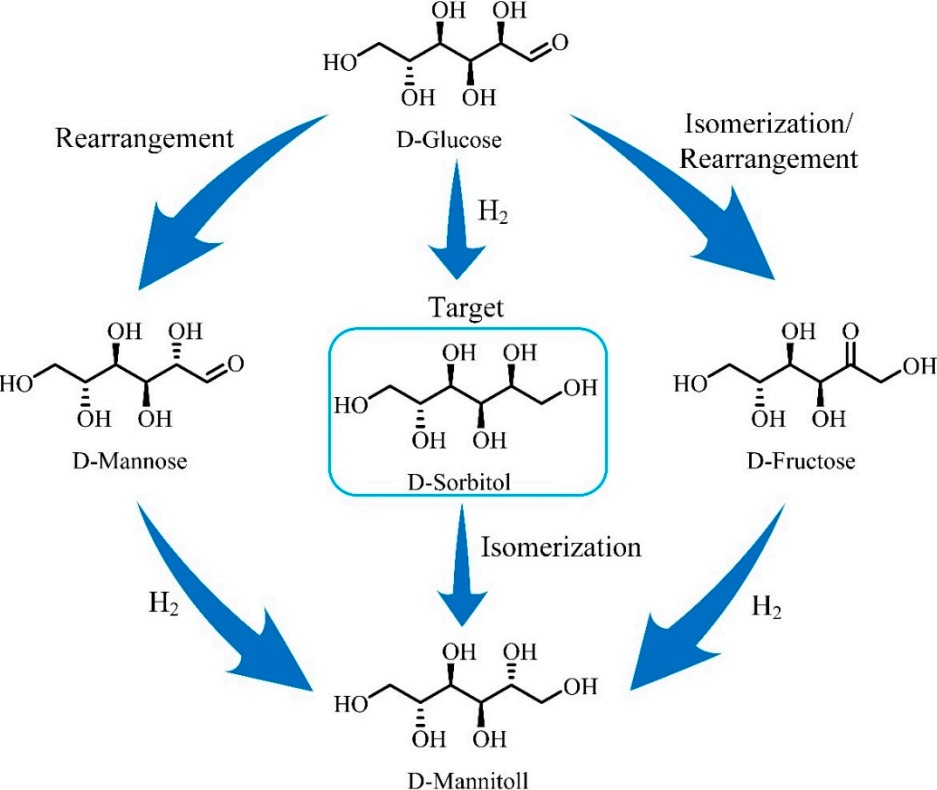

**Scheme 1.** Hydrogenation of glucose to sorbitol with a possible side way.

As reported by the literature, grafting a multifunctional shell, such as $ZrO_2$ [8], or introducing nitrogen atoms [9], the dispersion of Ru nanoparticles (Ru NPs) on the support could be promoted [8]. In our previous work, the amino poly (styrene-*co*-maleic) polymer (ASMA) has verified its plausibility to coordinate with the electro-deficient noble metal ions firmly [32]. The discrete functional groups with -OH and -$NH_2$ groups could precisely serve as the target spot to conjugate Ru cations. After being activated with hydrogen, Ru NPs can be formed and embedded into the polymer surroundings and strongly bounded through the interactions [33–36]; thus, the agglomeration and leaching of the active sites can be excluded.

In addition, the most currently used catalysts in glucose hydrogenation can be classified into the powder catalysts [4,5,8], which bring forth a significant pressure drop with liquid flow passing in the column, not suitable in scale-up applications [37]. Building the catalyst with a firm-shaped subject is necessary for improving the productivity. Through impregnation on different supports, such as commercial active carbon support [20], carbon nanotubes [38], etc. [39], the resulted shaped samples present its feasibility because the size of the resulted pellet or granules could decrease the pressure drop resulted from the hydrodynamics, which could guarantee the access to the continuous

flow process [40]. Among all these shaped subjects, the sphere-like γ-Al$_2$O$_3$ pellets have verified its superiority attributed to the high surface area [41], strong mechanical solidity [42], and a defined diameter [32], which is frequently used in trickle bed reactors [40,43].

Based on the analysis above, to construct a core-shell-like structure with a rigid-shaped core encapsulating the multifunctional shell could satisfy these demands. With the high dispersion of Ru induced by the electron-donating groups on the multifunctional shell, the Ru NPs can be homogenously dispersed on the external surface with the outward direction [32]. At the same time, the rigid core could serve as the shaped subject to bear the harsh conditions, especially to avoid the complex hydrodynamics influence and minimize the pressure drop in the bed column due to the huddle packing of powder catalysts in the bed [17,44,45].

Herein, we proposed an easy preparation method for the core-shell-like Ru catalyst through embedding the Ru nanoparticles (Ru NPs) on the polymer encapsulating γ-Al$_2$O$_3$ pellet supports. During the preparation procedure, with direct immerging into the polymer precursor solution, the polymer shell with abundant -NH$_2$ and -OH sites can be formed on the surface of the macro-porous γ-Al$_2$O$_3$ pellets. Then, the Ru ions were homogenously implanted into the supports' surface by using these electro-donating groups located inside the ASMA polymer subject as a target spot through coordination bonds. After being activated under hydrogen environment, the resulted Ru NPs can be solidly embedded onto the surface of supports. In addition, the high mechanical strength of γ-Al$_2$O$_3$ pellet could guarantee the resulted sphere Ru catalysts to be a suitable candidate in the continuous reaction system under harsh conditions. Therefore, the resulted sphere Ru catalyst was followed by applying it in the hydrogenation of glucose under both batch reactor and a trickle bed reactor to yield sorbitol efficiently. A mechanism on the structure-activity relationship of this intriguing catalyst has also been established.

## 2. Results and Discussion

### 2.1. Crystallinity and Textural Properties

The crystallinity of the resulted samples can be determined by powder XRD measurements. As shown in Figure 1, the obscure peaks in XRD patterns indicate the low crystalline nature of the macro porous γ-Al$_2$O$_3$, as well as the one encapsulated with ASMA. In the sample of γ-Al$_2$O$_3$@ASMA, the presence of typical γ-Al$_2$O$_3$ planes (JCPDS 75-0921) can be observed. The diffraction peaks at 14.18, 28.02, 38.06, and 49.40° can be scripted to the (004), (102), (122), and (203) planes to further deteriorate the crystallinity of the samples. In addition, planes of the γ-Al$_2$O$_3$ [46]. It is worth noting that, after encapsulation, the modification process could not be found after impregnating with 5 wt% Ru, only a small diffraction peak marked with "*", which is assigned to the (101) plane of Ru NPs [9,47], can be found. Through calculation by Scherrer equation, the crystallite size of Ru NPs from (101) plane is only 4.54 and 4.57 nm in sample 5%Ru/γ-Al$_2$O$_3$@ASMA and its counterpart 5%Ru/γ-Al$_2$O$_3$, respectively.

The N$_2$ absorption-desorption isotherms of 5%Ru/γ-Al$_2$O$_3$@ASMA and Ru/Al$_2$O$_3$ samples are depicted in Figure 2a, presenting the type V hysteresis loop, according to the IUPAC. These results indicate that the encapsulation and impregnation process would not block the macropores and mesopores in the original Al$_2$O$_3$ pellets. However, it is obvious to observe that accompanied with the encapsulation process, the loops tend to shrink as shown by the BET results in the supporting information part. The BET surface area would decrease sharply from 181.93 to 119.44 m$^2$/g, which may result from the high fluidity of the polymer and deteriorate the adsorption property of the resulted samples (see Table S1). After immobilizing with Ru NPs, both of the supports, before and after encapsulation with ASMA, would drop, which can be dedicated to the formation of metal clusters in the macropores of the support. Through the investigation of the pore distributions based on the BJH method (Figure 3b), the pore diameter distribution at mesopores section (2–50 nm) of the sample without ASMA encapsulation would turn to obscure, while, intriguing, the mesopores proportion

can unexpectedly survive. This may indicate that the assembling of Ru NPs mainly on the external surface is firmly bounded by the polymer shell through the -NH$_2$ and -OH groups. It could restrain the random deposition of Ru NPs clusters in the support's pores [32].

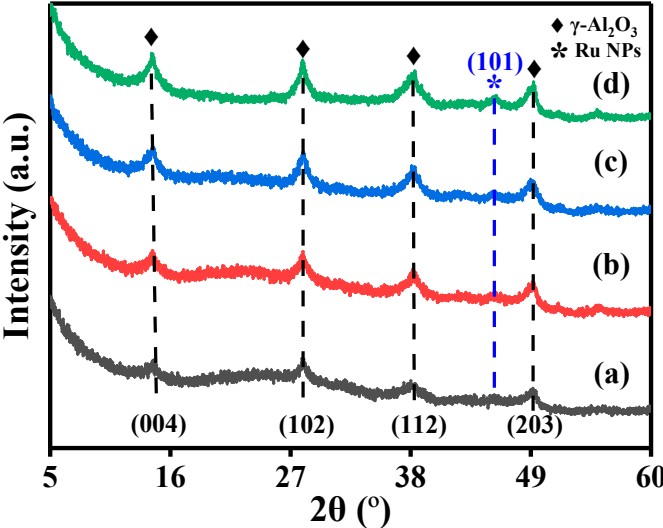

**Figure 1.** XRD pattens of (**a**) γ-Al$_2$O$_3$, (**b**) γ-Al$_2$O$_3$@ASMA, (**c**) 5%Ru/γ-Al$_2$O$_3$@ASMA, and (**d**) 5%Ru/γ-Al$_2$O$_3$.

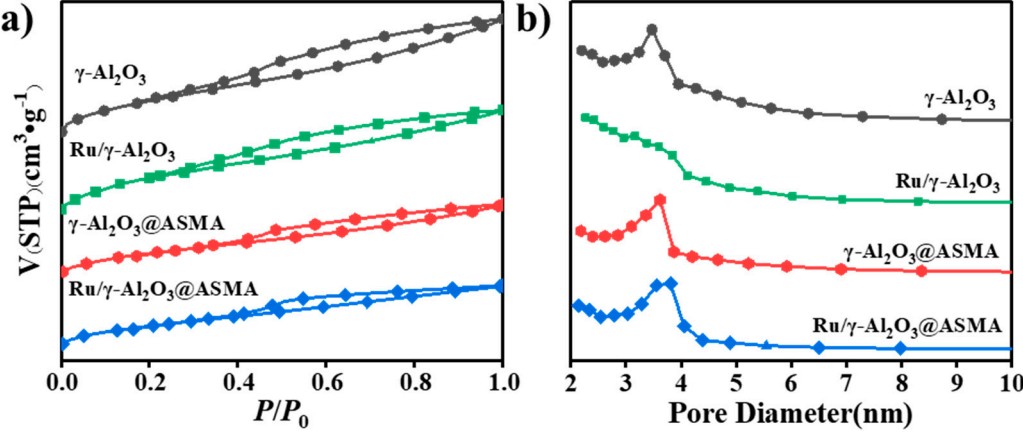

**Figure 2.** (**a**) N$_2$-BET and (**b**) pore size distributions patterns of catalysts and intermediates.

*2.2. Morphology and Microstructure*

In order to confirm the influence of the encapsulation process on the γ-Al$_2$O$_3$ sample, the morphology of the γ-Al$_2$O$_3$ support before and after modification has been investigated through the SEM. It can be observed that the macropores in the γ-Al$_2$O$_3$ support have no change after encapsulating with ASMA. The rambling scattering of the Al$_2$O$_3$ particles has been kept, and no preconceived binding from the cross-linked polymer is observed. After impregnating with Ru NPs, the resulted 5%Ru/γ-Al$_2$O$_3$@ASMA catalyst still maintains the typical morphology of the γ-Al$_2$O$_3$ support. This result could provide the information that no matter what the process of encapsulation and impregnation is, these post-treatments could not influence the morphological features of the initial γ-Al$_2$O$_3$ support. However, the adequate macro and mesopores of the support could bring enough channel species for the formation of Ru NPs clusters after impregnation. It could be observed from the TEM result of 5%Ru/γ-Al$_2$O$_3$@ASMA that the ruthenium metal cluster particles are evenly distributed on the support (Figure 3d). The typical TEM micrograph and the particle-size distribution

histogram of Ru nanoparticles on γ-Al$_2$O$_3$@ASMA support are shown in Figure 3d, presenting a mean Ru particle size of ca. 7.80 nm, which almost agrees with the above XRD result that no obvious Ru diffraction peaks can be observed. This result could confirm the high distribution of the Ru NPs on the surface of the γ-Al$_2$O$_3$@ASMA support, which would benefit the hydrogenation of glucose. In addition, combining the result of a sharp decline of BET surfaces and constant pore size, the unchanged morphology could further verify that the ASMA only covers the γ-Al$_2$O$_3$ support without shrinkage of the catalyst pore [48]. However, it should be noted that the function of the ASMA layer is still unclear, in order to distinguish its role in detail, more in-depth characterizations should be applied.

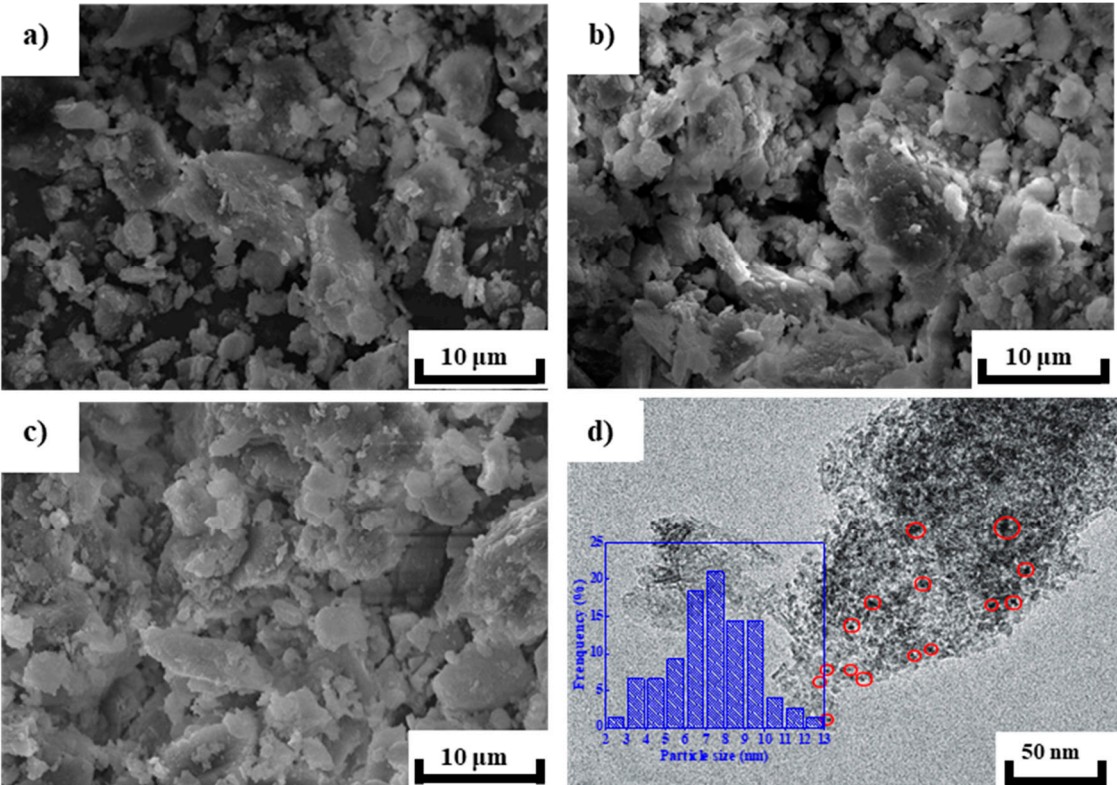

**Figure 3.** SEM images of (**a**) γ-Al$_2$O$_3$, (**b**) γ-Al$_2$O$_3$@ASMA, and (**c**) 5%Ru/γ-Al$_2$O$_3$@ASMA; (**d**) TEM image of 5%Ru/γ-Al$_2$O$_3$@ASMA. The insert represents the histogram of the corresponding Ru NPs size distribution in 5%Ru/γ-Al$_2$O$_3$@ASMA.

### 2.3. Surface Characterization of the Ru Catalyst

The presence of the electron-donating group including -OH and -NH$_2$ groups on the ASMA shell can be confirmed through FTIR spectra. As shown in Figure 4, the broad weak peak at 3033 cm$^{-1}$ may be ascribed to the bonded hydroxyl (-OH) or amine groups (-NH$_2$) on the ASMA shell. In addition, the peaks at 1600, 1500, and 1493 cm$^{-1}$ can be attributed to the skeletal vibration of benzene ring [49]. The peaks at 699 and 758 cm$^{-1}$ resulted from the out-of-plane bending vibration of C–H in benzene ring [50]. The bands around 1710 and 1778 cm$^{-1}$ were due to the symmetrical and asymmetrical stretching vibration of C = O groups in maleic anhydride units, respectively. The presence of para-substituted benzene can be detected through the acute medium peak at 1450 cm$^{-1}$ [32].

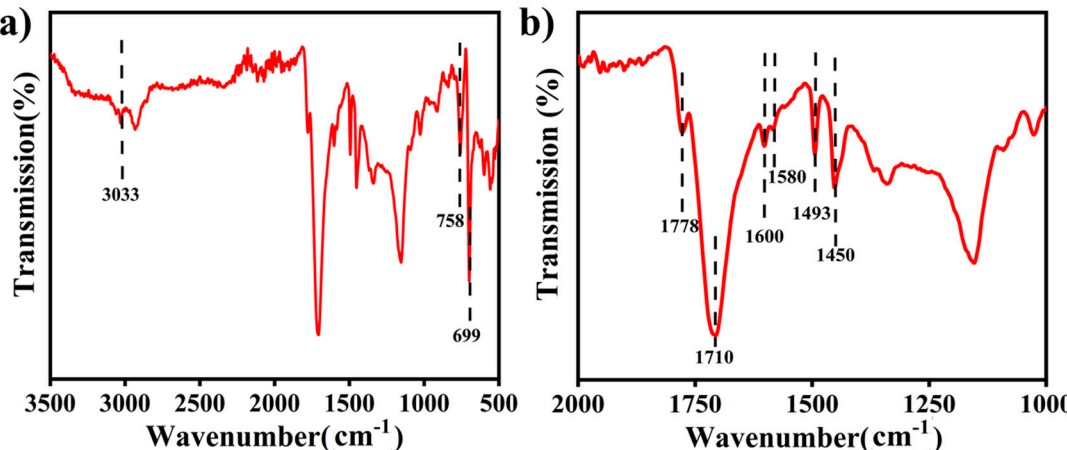

**Figure 4.** FT-IR diagram of polymer shell without the $\gamma$-Al$_2$O$_3$ pellet in transmission mode. (**a**) 3500–500 cm$^{-1}$; (**b**) 2000–1000 cm$^{-1}$.

In order to investigate the interaction between the presence of the electron-donating groups on the support surface and the Ru active sites, the XPS measurement was applied in this work. The XPS survey scans for the catalyst 5%Ru/$\gamma$-Al$_2$O$_3$@ASMA and its counterpart 5%Ru/Al$_2$O$_3$ can be found in Supplementary Information. Figure 5 presents the XPS results in the spectral regions of the Ru $3d$, Ru $3p$, Al $2p$, and O $1s$ in both of these two catalysts. An overall decline of the signal intensity of the peaks in 5%Ru/$\gamma$-Al$_2$O$_3$@ASMA could be observed. It can be explained as the signal block on the catalyst surface due to the encapsulation of the ASMA polymer layer, being a solid proof to the core-shell structure of the resulted catalyst. Moreover, the typical XPS peaks of Ru $3d_{3/2}$ and Ru $3d_{5/2}$ at 284.5 and 281.1 eV exist in Figure 5a, confirming that the Ru species on both of these two catalysts are in the metallic state. Since the Ru $3d_{3/2}$ peaks are overlapped by the C $1s$ level [51], it is only the peak at 281.1 eV assigned to the Ru $3d_{5/2}$ level that can be applied to provide more information. It is intriguing to find that the peak intensity assigned to Ru $3d_{5/2}$ in 5%Ru/$\gamma$-Al$_2$O$_3$ is much lower than that of its counterpart in 5%Ru/$\gamma$-Al$_2$O$_3$@ASMA, which is extraordinary to the signal block effect of the polymer encapsulation. Therefore, it can only be attributed to the much higher adsorption amount of the Ru species in 5%Ru/$\gamma$-Al$_2$O$_3$@ASMA, indicating the better impregnation property due to the presence of ASMA polymer layer.

In addition, as presented in the result of Ru $3p$ in Figure 5b, both the samples of 5%Ru/Al$_2$O$_3$ and 5%Ru/$\gamma$-Al$_2$O$_3$@ASMA display the typical high binding energy values at 462.1eV of the Ru $3p_{3/2}$ orbital, which confirm the presence of the mixture RuO$_x$ and Ru species in the catalysts [52]. However, the unchanged binding energy of these catalysts neglects the influence of ASMA polymer to the Ru sites. This result could verify that the polymer layer only plays the role of attracting the Ru$^{3+}$ precursor before activation. In particular, the intensity decline of the Al $2p$ XPS spectra at 74.3 eV in 5%Ru/$\gamma$-Al$_2$O$_3$ and 5%Ru/$\gamma$-Al$_2$O$_3$@ASMA also manifests the block of detection of the target Al atoms on the surface (Figure 5c) [53], which significantly indicates the change of ratio of Al and C atoms due to the introduction of ASMA polymer.

With regard to the presence of RuOx, it is critical in this work to distinguish the function of the support and the polymer, and also to investigate the XPS spectra in the O $1s$ region. As shown in Figure 5d, both of these two catalysts presented three peaks which can be attributed to the lattice oxygen (O$_{latt}$) at 531.6 eV, adsorbed OH group at 532.6 eV, and Al-O at 530.5 eV [54,55]. The absence of the carbonyl oxygen at 530.8 eV, resulted from the existence of C = O groups in maleic anhydride units, while confirmed by the FTIR result, could be attributed to the overlapping of the O $1s$ peak of Al-O [56]. However, the slight change in binding energy values could manifest that the encapsulation process could bring out no influence to the support as well as to the Ru active sites.

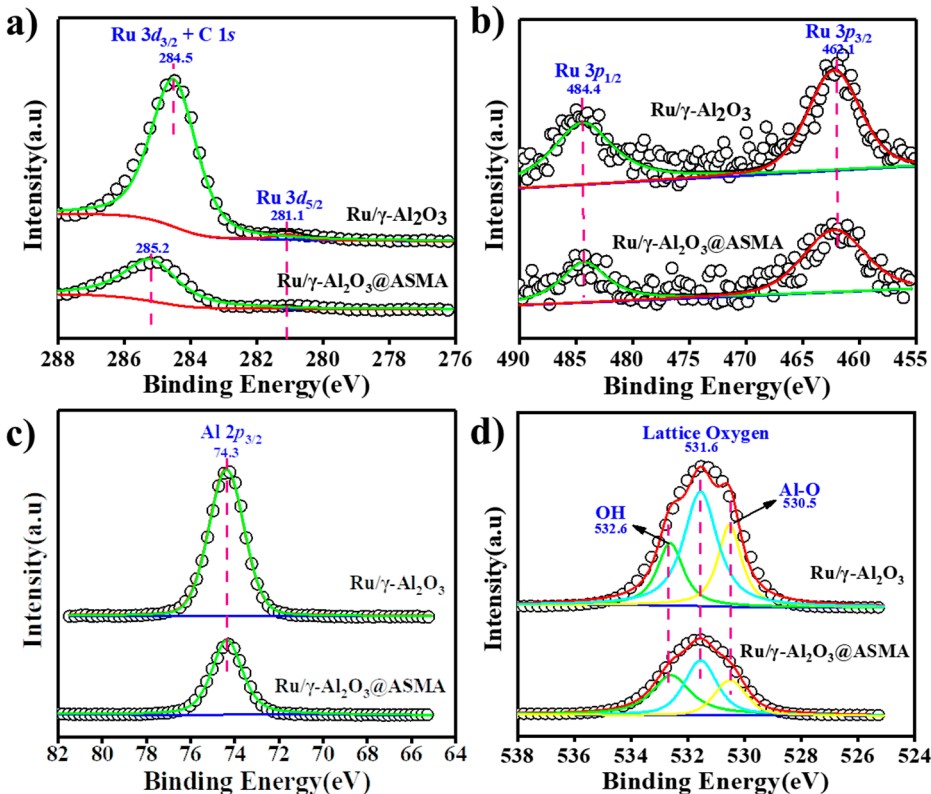

**Figure 5.** XPS spectra of (**a**) Ru *3d*, (**b**) Ru *3p*, (**c**) Al *2p*, and (**d**) O 1*s* for the 5% Ru/γ-Al₂O₃ and 5% Ru/γ-Al₂O₃@ASMA.

*2.4. Catalytic Activity*

Before testing with a catalytic reaction, the ratio of ASMA polymer and γ-Al₂O₃ as well as the thermal stability of the 5%Ru/γ-Al₂O₃@ASMA pellet was investigated by TGA, and a three-step weight loss can be observed. It is worth noting that, the weight loss of the sample should mainly be attributed to the decomposition of the ASMA polymer. In order to clarify the decomposition process, the curve subtracted with the solidified γ-Al₂O₃ pellet has also been investigated with the pure Ru impregnated pristine ASMA polymer. As shown in Figure S2, the first weight loss at about 150 °C can be attributed to the removal of water molecules trapped in the pores. The following loss in the range of 150 to 300 °C is due to the dehydration of the hydroxyl groups on the surface and ASMA polymers. After 300 °C, the decomposition process of the ASMA polymer grafting shell can be proceeded, which presents a total weight loss of 21.17% to the Ru catalyst as-prepared and 82.30% to the sample subjecting the γ-Al₂O₃ pellet subject, determining the ratio of ASMA polymer and γ-Al₂O₃ to be almost 1:2.88 [48]. This result indicates a high thermal stability of the 5%Ru/γ-Al₂O₃@ASMA catalyst, which can be applied in the hydrogenation of glucose.

Therefore, the catalytic activities of the resulted 5%Ru/γ-Al₂O₃@ASMA catalyst were evaluated for the hydrogenation of glucose under a batch reactor [2,15]. First, the influence of the reaction temperature to the activity and selectivity to sorbitol was investigated by proceeding the reaction in the range of 90–130 °C under 5 MPa H₂ environment for 2 h (Figure 6a). With the rise of the temperature, the conversion of glucose sharply increases from 35.98% to almost completely consumed. However, the selectivity to the sorbitol is dominated by the temperature rise. The high temperature, for example at 130 °C, would result in more side-reactions such as glucose coking [2], glucose isomerization [26], sorbitol decomposition [57], etc. [17], which would lead to the deterioration to the sorbitol selectivity. The yield of target sorbitol declined to only 59.64%, while the optimal temperature can be found at 120 °C with the target yield of 95.19%.

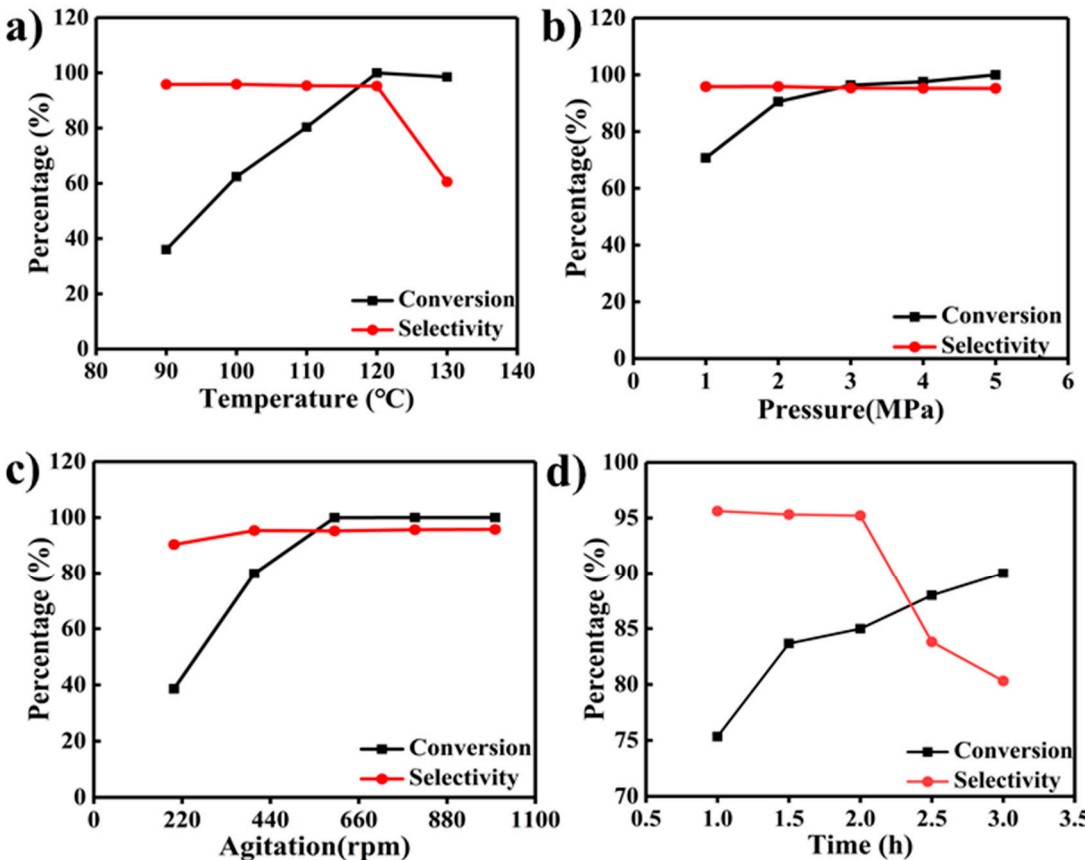

**Figure 6.** Catalytic activity and selectivity test with different conditions: (**a**) Temperature, (**b**) pressure, (**c**) agitation, and (**d**) time.

Figure 6b illustrates the relationship between hydrogen pressure to the glucose conversion and sorbitol selectivity from 1 to 5 MPa. Different from the result shown in Figure 6a, the gentle rise of the sorbitol yield from 2 to 5 MPa (from 86.74% to 95.19%) also indicates the high activity of the resulted core-shell catalyst, which decreases the sensitivity to the hydrogen pressure. The best reaction result can be acquired at 5 MPa. It should be noted that the agitation rate also plays a critical role in the glucose hydrogenation reaction, especially if it is a pseudo-first order reaction dependence with respect to the glucose [13]. The mass transfer should not be ignored due to the limited hydrogen diffusion in the tri-phase reaction system. From Figure 6c, it could be observed that the glucose conversion and sorbitol selectivity reach the demanded value after the agitation rate is more than 600 rad/min. However, the potential risk and energy consumption should also be considered in this work. Therefore, the optimal reaction conditions can be set at 600 rad/min with 5 MPa $H_2$.

In order to test the influence of reaction time, especially to the reaction pathway of glucose hydrogenation, the experiment was carried out at 600 rad/min with 5 MPa $H_2$ under 120 °C for different reaction times. As shown in Figure 6d, the glucose conversion rises with increasing the reaction time, while the sorbitol selectivity presents a volcano tendency. The glucose raw material could be completely consumed after only 2 h, with the optimal sorbitol selectivity of 95.21%. However, the selectivity unexpectedly turns to the inflection point which brings forth only 54.31% selectivity after reacting for 3 h. The sharp decline of selectivity could be attributed to the further degradation of the yielded sorbitol under hyper-thermal conditions [2]. Mannitol, 1,3-propanediol, 1,4-butanediol, glycerol, even 5-HMF can be finally formed due to the unavoidable side reactions [27,57,58].

### 2.5. Reusability and Peer Comparison

After the reaction, the catalyst was separated from the solution phase, washed with deionized water, and dried at 110 °C overnight for the next run. From Figure 7a, no significant loss in sorbitol yield was observed up to five times, which suggested that Ru was not leached during the reaction. For the comparison, the 5%Ru/$\gamma$-Al$_2$O$_3$ counterpart was also tested as a catalyst in the same reaction procedure, which cannot maintain the similar high sorbitol yield only after two times. Based on this result and the characterizations beforementioned, it could be explained as the embedded Ru NPs firmly confined by the polymer environment [36]. The polymer shell encapsulated on the $\gamma$-Al$_2$O$_3$ plays a critical role in maintaining the Ru NPs without leaching.

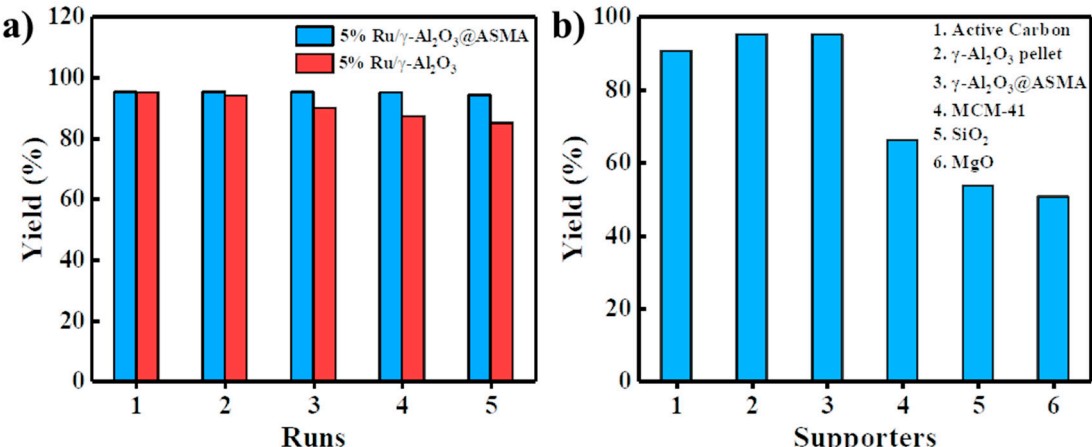

**Figure 7.** (**a**) Catalytic reusability of 5%Ru/$\gamma$-Al$_2$O$_3$@ASMA and 5%Ru/$\gamma$-Al$_2$O$_3$; (**b**) peer comparison with different supports. Reaction condition: Catalyst: 2 g; temperature: 120 °C, H$_2$ pressure: 5 MPa; agitation rate: 600 rad/min; solution concentration: 20 wt% aqueous glucose, 50 g; reaction time: 2 h.

In addition, other peer supports, including active carbon, MCM-41, SiO$_2$, and MgO have also been selected to prepare the 5% Ru catalysts in this work. Illustrated in Figure 7b, under the same reaction conditions, 5%Ru/$\gamma$-Al$_2$O$_3$@ASMA has verified to be the best catalyst among all these peer catalysts. Compared with the catalysts reported in the recent literature, such as Ru-ZrO$_2$-SBA-15 [8], Raney nickel catalysts [23], Pt/C catalyst [3], Ni-supported catalyst [1], etc. [10,27,56], the 5%Ru/$\gamma$-Al$_2$O$_3$@ASMA catalyst herein surpasses them in the aspect of temperature [3], reaction hour [8], and sorbitol yield [23].

### 2.6. Continuous Test

Generally, the flow process with a continuous reactor could facilitate the reaction to be operated for a long period at a large scale, which would expand the scalability to the industrial purpose with high efficiency [44]. The presence of the firm $\gamma$-Al$_2$O$_3$ pellet core in the beforementioned catalyst 5%Ru/$\gamma$-Al$_2$O$_3$@ASMA brings forth the access to be applied in the flow process by decreasing the pressure drop [17,57]. Considering the high activity of the catalyst, the as prepared sample is placed in a self-made trickle bed for the continuous production of glucose hydrogenation. According to the previous literature and the practical experience [17,20,22,38,44,45,57,59], through investigating the catalyst under a long-term test at 110 °C, 2.5 MPa, with the injection flow rate ($Q_{injection}$) of 5 wt%, glucose aqueous solution at a rate of 1 mL/min and flow rate of hydrogen ($Q_{H2}$) at a rate of 10 mL/min, as shown in Figure 8, it can be confirmed that the hydrogenation can still stay positive even after 1000 h, which presents no less than 94% selectivity to sorbitol accompanied with almost 90% consumption of glucose. The supreme catalytic property and stability are mainly attributed to the strong binding of Ru surrounded with the electron-donating groups on the polymer shell and the inherited large pores from the inner $\gamma$-Al$_2$O$_3$ pellet core, which guarantee that the catalyst as prepared in this work can be an excellent candidate in the efficient production of sorbitol through glucose hydrogenation.

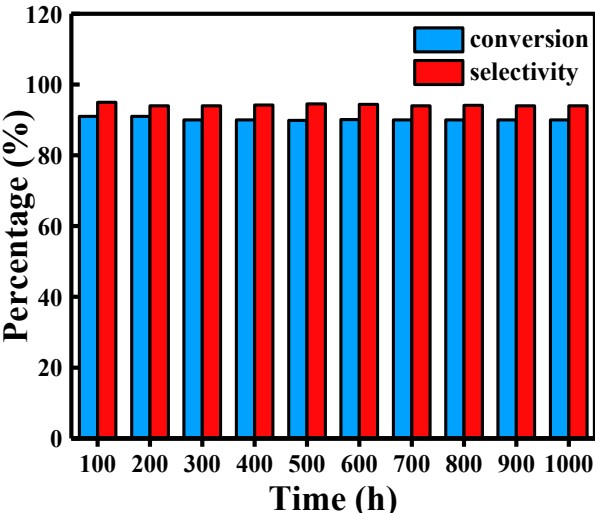

**Figure 8.** Catalyst life inspection in the trickle bed reactor. Reaction condition: Temperature: 110 °C, $H_2$ pressure: 2.5 MPa; $Q_{injection}$ = 1 mL/min; $Q_{H2}$ = 10 mL/min; solution concentration: 5 wt% aqueous glucose.

## 3. Catalytic Mechanism Discussion

The maleic anhydride unit is highly reactive towards surface hydroxyl groups [48], which could serve as the external shell enabling the uniformly dispersion of the polymer precursor on the hydroxyl rich $\gamma$-$Al_2O_3$ core. Through the directly inductive effect of the external hydroxyl group, the ASMA polymer shell can be grafted on the surface of $\gamma$-$Al_2O_3$ densely. During impregnation, the outmost -$NH_2$ and -OH groups play a critical role as a nucleophilic target spot to attract more dissociative Ru cations to be accumulated outward separately on the surface [44]. After activating with hydrogen, the Ru NPs can be embedded on the hierarchical porous $\gamma$-$Al_2O_3$ pellet firmly [32]. The highly dispersed Ru could accelerate the hydrogenation of glucose to sorbitol rather than the side reactions such as isomerization [26], carbonization [60], and cleavage reactions [10,57]. During the hydrogenation, the introduced $H_2$ was first adsorbed on the uniformly scattering Ru catalytic sites to form the active hydrogen species. After the coalition with glucose molecules, the active hydrogen would immediately attack the carbonyl groups of the reactant [6], and the target sorbitol can be formed and desorbed into the mixture solution (see Scheme 2). Due to the strong hydrogen bonds attributed from the external -OH and -$NH_2$ on the ASMA shell, different from the pure van de Waals force between Ru NPs and $\gamma$-$Al_2O_3$ pellet, it could much stronger bind the Ru NPs on the support. The strong interaction between Ru NPs and ASMA that encapsulated $\gamma$-$Al_2O_3$ would further promote the recyclability of the catalyst. Moreover, the resulted catalyst can be located in the shaped sphere catalysts which is suitable under harsh reaction conditions, such as being applied in the bed reactors, due to the mechanically solid $\gamma$-$Al_2O_3$ pellet core. It is effective for the application in this work to choose the trickle bed reactor for the highly efficient production of sorbitol under environmentally friendly reaction conditions [39,43].

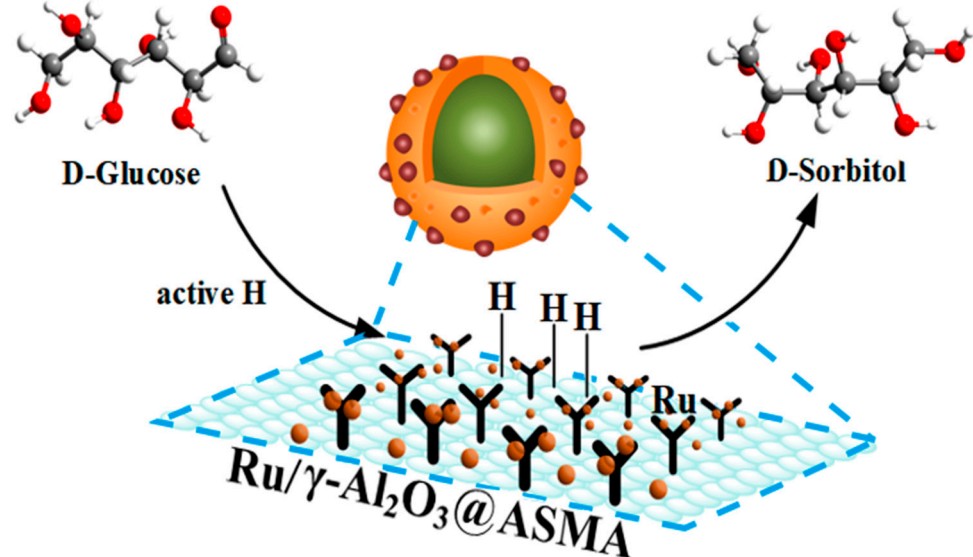

**Scheme 2.** The hydrogenation route from glucose to sorbitol with the 5%Ru/γ-Al₂O₃@ASMA pellet as catalysts. The insert core-shell pellet represents the 5%Ru/γ-Al₂O₃@ASMA pellet.

## 4. Materials and Methods

### 4.1. Materials

The amino poly (styrene-*co*-maleic) polymer (ASMA) encapsulated γ-Al₂O₃ pellets supports were synthesized according to our previous work [32]. First, the γ-Al₂O₃ pellets (4 mm) were purchased from Henan Soar Environment Technology Company (Zhengzhou, China). Maleic acid (AR, Aladdin, Shanghai, China), Styrene (CP, Zhuyue, Shanghai, China), and divinylbenzene (AR, Alfa Aesar, Shanghai, China) were applied as ASMA polymer precursors; benzoyl peroxide (AR, Aladdin, Shanghai, China) was used as the initiator in the process of polymerization; methanol (AR, Yonghua, Changshu, China) was used as the pore-forming agent. All these chemicals, including aqueous ammonia (AR, Linfeng, Shanghai, China) and hydrochloric acid (AR, Sinopharm, Shanghai, China) and others, were provided by local suppliers. During the investigation of hydrogenation, glucose (AR) was purchased from Sinopharm Chemical Reagent Company (Shanghai, China) and ruthenium chloride (GR) was used as a Ru source and acquired from Aladdin Biochemical Technology Company (Shanghai, China). The materials in this work are used without further purification. In addition, more information about the other peer catalysts, including 5% Ru impregnated active carbon, MCM-41, SiO₂, and MgO can be found in the Supplementary Materials.

### 4.2. Catalyst Preparation

First, γ-Al₂O₃ pellets were sonicated in water for 30 min to remove the surface impurity and dried at 105 °C overnight for further usage, and then immersed into the polymerization solution containing styrene, maleic anhydride, divinylbenzene, and benzoyl peroxide which were mixed in the molar ratio of 200:200:20:1, respectively. Methanol was added as a pore-forming agent during the process. The pellets were immerged and kept for 3 h. Afterwards, the pellets were filtrated from the polymerization solution and drained completely until no liquid drops failed. The post-treated pellets were purged with N₂ gas in the oven and heated through the temperature-programmed heating process. Poly (styrene-*co*-maleic anhydride) polymer (SMA) was produced and encapsulated on the surface of γ-Al₂O₃. The resulted milky firm polymer encapsulated pellets were transferred into an ammonia aqueous solution under 45 °C for ammonization. After being neutralized with 10% hydrochloric acid, the target core-shell-like ASMA encapsulated γ-Al₂O₃ supports (γ-Al₂O₃@ASMA) were obtained. The Ru species of 5% by weight were introduced into the γ-Al₂O₃@ASMA support through the

conventional wet impregnation method. The 2.0 g γ-Al$_2$O$_3$@ASMA support was transferred into the round bottom flask with the prestaging RuCl$_3$ ethanol solution of 0.2053 g RuCl$_3$ and refluxed for 12 h with continuous stirring at 78 °C. The impregnation process was finished when the solvent fully evaporated. The resulted samples were collected, washed, and dried at 105 °C overnight. Before being applied in the hydrogenation reaction, the as-prepared 5%Ru/γ-Al$_2$O$_3$@ASMA precursor was placed in a batch reactor filled with 5 MPa H$_2$ at 120 °C for 3 h. After washing with water to remove the residue chlorine ions and dried, the target 5%Ru/γ-Al$_2$O$_3$@ASMA pellet catalyst can be achieved. As for comparison, the traditional 5%Ru/γ-Al$_2$O$_3$ pellet catalyst was also prepared with the wet impregnation method, which is almost similar to the preparation of the 5%Ru/γ-Al$_2$O$_3$@ASMA pellet catalyst but without the encapsulation process.

### 4.3. Catalyst Characterization

The morphology of the catalyst and Ru NPs were measured with scanning electron microscopy (SEM) and transmission electron microscopy (TEM). Before testing, all the samples were crushed into mesh-like powders by grinding in a crystal mortar. The SEM micrographs were acquired using a Hitachi S-4800 (Hitachi, Tokyo, Japan) electron microscope working at 200 kV. The TEM results were proceeded through a JEOL JEM-2010 microscope with a 200 kV accelerating voltage. During the TEM analysis process, a few milligrams of sample were dispersed into 2 mL ethanol with ultrasonication for 30 min and dropped on a copper grid covered with a 300-mesh holey layer carbon film.

The crystallinity of the samples after being crushed into mesh-like powders were determined by X-ray diffraction (XRD) patterns and recorded on a Rigaku Smartlab TM 9KW diffractometer (Rigaku, Japan) equipped with a Cu Kα radiation (λ = 1.542 Å), operating at 40 kV and 100 mA, in the angle range of 2θ = 5–60° with a step size of 0.03. As for confirming the thermo stability and polymer/γ-Al$_2$O$_3$ ratio of the resulted catalyst, a STA 449F3 thermogravimetric analyzer (TGA) (NETZSCH, Berlin, Germany) at a heating rate of 10 °C/min from 30 to 800 °C under N$_2$ flow was used in this work.

X-ray photoelectron spectroscopy (XPS) measurements for the crushed catalyst powders were performed by a K-Alpha spectrometer (Thermo Scientific, New York, NY, USA) with a monochromatic Al-Kα ray source under ultra-high vacuum conditions. In order to exclude the external contamination, the samples were pretreated under an argon flow for 2 h, while they were measured directly without further activation. Binding energy (BE) values were calibrated referenced to the C 1*s* peak (284.8 eV) during data processing of the XPS spectra. The peaks decompositions assigned to the Ru 3*d*, Ru 3*p*, Al 2*p*, and O 1*s* binding energy values were analyzed by applying mixed Gaussian-Lorentzian profiles and a Shirley nonlinear sigmoid-type baseline fitting of varying proportions (30–80%).

Textural properties were characterized through the nitrogen sorption-desorption isotherms at −196 °C using an BK122W analyzer (JWGB Sci and Tech Ltd., Beijing, China). During this process, it is not necessary to crush the sample as beforementioned due to the analysis request. Before analysis, the pellet-like samples were degassed under a vacuum for 6 h at 150 °C. The specific surface areas (SSA) and porosity were achieved by the calculation of BET and BJH methods, respectively.

The determination of the existence of the electron-donating group including -OH and -NH$_2$ groups on the ASMA polymer were performed through the Fourier transform infrared (FTIR) spectra, which were analyzed with an Agilent Cary 660 spectrometer (Agilent, Santa Clara, CA, USA) in the range of 400–4000 cm$^{-1}$ in transmission mode. It should be noted that, for clarification, only the pristine ASMA polymer shell, subtracting the γ-Al$_2$O$_3$ pellet, were investigated in this work. The samples were prepared by mixing the products with KBr and pressing into a compact pellet.

### 4.4. Catalytic Hydrogenation of Glucose to Sorbitol

The hydrogenation of glucose was performed in a 100 mL stainless autoclave under vigorous stirring. Following the loading of the autoclave with 2 g of catalyst and 50 g of a 20 wt% glucose solution, the reactor was purged with hydrogen four times to remove air. The reactor was heated in the temperature range of 90–130 °C, and then kept at the preset hydrogen pressure from 1 to 5 MPa.

During hydrogenation, the initial stirring rate was kept at the desired value (200–1000 rad/min). After the reaction, the solid catalyst was separated, washed, and dried under vacuum drying at 100 °C for the next run. The residue solution was immediately filtered with a needle-drum membrane filter of 0.45 μm and analyzed by high-performance liquid chromatography (HPLC) (Shimadzu, Japan) with refractive index (RI) detection. The HPLC column applied in this work was a SUGAR SC-1011 column (Shodex, Japan) at 80 °C with a flow of 1 mL/min using deionized water as the mobile phase [13]. Glucose conversion and sorbitol selectivity were calculated using Equations (1) and (2) as follows:

$$\text{Conversion} \ = \frac{[\text{Glucose}]_i \ - \ [\text{Glucose}]_f}{[\text{Glucose}]_i} \times 100\% \tag{1}$$

$$\text{Selectivity} \ = \frac{[\text{Sorbitol}]}{[\text{Glucose}]_i \ - \ [\text{Glucose}]_f} \times 100\% \tag{2}$$

where $[\text{Glucose}]_i$ and $[\text{Glucose}]_f$ represented the initial and final molar quantity of glucose, separately. $[\text{Sorbitol}]$ represented the molar quantity of sorbitol as detected.

### 4.5. Continuous Reaction

It is worth noting to bear in mind that the $\gamma$-Al$_2$O$_3$ pellets should be almost uniformly shaped with 4 mm stable sphere particles; the resulted catalysts are also very suitable to be applied under the harsh reaction conditions in highly efficient continuous reactors [39]. In order to promote the production efficiency and glucose hydrogenation productivity with the resulted 5%Ru/$\gamma$-Al$_2$O$_3$@ASMA pellets as catalysts, the reaction was investigated under a typical trickle bed reactor (see Figure 9). The hydrogenation of glucose was conducted in a self-made trickle bed reactor using a 580 mm in length and 20 mm in diameter stainless steel column directly connecting to a LC-P100 high-pressure pump (Wufeng, Shanghai, China) for liquid recycle (1 mL/min, 5 wt% glucose aqueous solution).

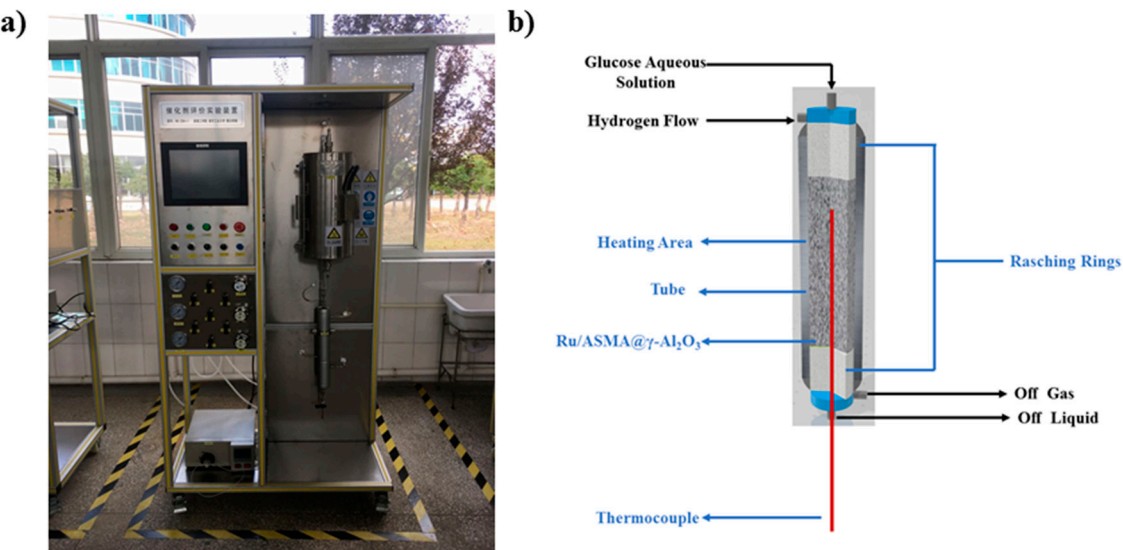

**Figure 9.** (**a**) Trickle bed used in this work; (**b**) reactor column filling and feedstock scheme.

To avoid the escape of catalysts from the reactor bed, Rasching rings were placed on the bottom of the trickle column. Furthermore, the reactor bed was filled with 50 g 5%Ru/$\gamma$-Al$_2$O$_3$@ASMA pellets catalysts until the tip of thermocouple was merely invisible. The residue area was filled with Rasching rings for the uniform distribution of mass solution. During the reaction, the stainless-steel column was encased in an electrically heating jacket equipped with three thermocouples to control the reaction temperature at 110 °C, and the hydrogen pressure in the inside tank column was kept in

2.5 MPa, with a hydrogen flow of 10 mL/min, by a K9153F backpressure regulator (Xiongchuan, Jiangxi, China), which could be censored with a soap film flow. The outlet liquid aliquots were periodically collected through a cooling tube at every 100 h intervals and analyzed immediately with the HPLC descripted beforementioned.

## 5. Conclusions

In summary, a core-shell-like sphere ruthenium catalyst, named as 5%Ru/γ-Al$_2$O$_3$@ASMA, has been successfully synthesized in this work, and can be efficiently applied in the hydrogenation of glucose to sorbitol. Through the investigation on the structure-function relationship of this catalyst, it could be expounded that with the interaction of the amino poly (styrene-*co*-maleic) polymer shell and Ru NPs, the Ru active sites can be homogenously dispersed on the external surface of the catalyst firmly due to the inducing effect of the polymer shell. The Ru NPs can be embedded into the environment of polymer full of electron-donating groups, which strengthens the stability of catalyst without leaching or agglomeration. Furthermore, the solid sphere γ-Al$_2$O$_3$ core could play a critical role in decreasing the pressure drop in a trickle bed reactor. The high activity and selectivity of the target sorbitol guarantees the catalyst herein presenting a steadfast sorbitol yield of almost 90% both in the batch reactor and trickle bed reactor. This result indicates the potential feasibility of the core-shell-like catalyst in the efficient production of sorbitol.

**Supplementary Materials:** The following are available online at http://www.mdpi.com/2073-4344/10/9/1068/s1. Table S1: Brunauer-Emmett-Teller (BET) surface area, pore size, and pore volume of supports and catalysts; Figure S1: HR-XPS survey scans for 5%Ru/γ-Al$_2$O$_3$@ASMA and 5%Ru/γ-Al$_2$O$_3$; Figure S2: TGA of 5%Ru/γ-Al$_2$O$_3$@ASMA and 5%Ru/ASMA samples subtracting the γ-Al$_2$O$_3$ core.

**Author Contributions:** J.Z. (Jianliang Zhu) conceived and designed the experiments; X.Y. performed the TGA and TEM experiments and data analysis; J.Z. (Jing Zhao), W.W., J.L., and W.X. performed the experiments, analyzed the data, and composed the manuscript; Y.O., X.F., and Y.Y. checked, edited, and revised the manuscript; C.D. contributed to the reagents/program management. All authors have read and agreed to the published version of the manuscript.

**Funding:** This research received no external funding.

**Acknowledgments:** This work was self-supported by Zhu's laboratory in Nanjing Tech University.

**Conflicts of Interest:** The authors declare no conflict of interest.

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
