# Peer review of "Efficient Sorbitol Producing Process through Glucose Hydrogenation Catalyzed by Ru Supported Amino Poly (Styrene-co-Maleic) Polymer (ASMA) Encapsulated on γ-Al2O3"

_catalysts, doi:10.3390/catal10091068_

Round 1

Reviewer 1 Report

A core-shell like sphere ruthenium catalyst, named as 5%Ru/ASMA@γ-Al2O3, has been synthesized in this work. It can be efficiently applied in the hydrogenation of glucose to sorbitol. It was expounded that with the interaction of the amino poly (styrene-co-maleic) polymer shell and Ru NPs, the Ru active sites can be homogenously dispersed on the external surface of the catalyst firmly due to the inducing effect of the polymer shell. The high activity and selectivity of the target sorbitol guarantee the catalyst herein presenting a steadfast sorbitol yield of almost 90% both in batch reactor and trickle bed reactor. This result indicates the potential feasibility of the core-shell like catalyst in the efficient production of sorbitol. In these studies extensive diagnostic techniques (XPS, FTIR, SEM, TEM, XRD, BET, TGA) for catalyst characterization were used.

The research area is of interest to both scientists and engineers.

Generally, the quality of this paper is quite good, including quality of the figures. I can recommend it for publication after minor editorial corrections.

I have minor editorial comments:

  1. Lines 271: instead of "462,1eV" should be “462,1 eV”;
  2. In some places, spaces between the numbers and the signs "℃" should be removed; The signs "℃" used are strange.

Author Response

Dear Reviewer: I am gratitude to receive your reviewing comments and thanks for your sparing time to read my manuscript. Your comments could be the highly encouragement to our subsequent work in the future. In the following part, I am gladly to respond and answer your comments and suggestions. Comment 1: Lines 271: instead of "462,1eV" should be “462,1 eV”. Response 1: I agree with the comments of the reviewers and have already corrected it in the corresponding sections of the manuscript. Comment 2: In some places, spaces between the numbers and the signs "℃" should be removed; The signs "℃" used are strange. Response 2: I agree with the comments of the reviewers and have already corrected it in the corresponding sections of the manuscript.

Reviewer 2 Report

This manuscript describes the Ru/polymer catalyst formed on gamma-alumina for glucose hydrogenation. This catalyst has good mechanical and chemical stability as well as high activity. The results are interesting and can be published. However, significant information is lacking in this manuscript and should be provided.

(1) Title and abstract: the word "glucose" as substrate should be included.

(2) Expression of the catalyst: The core-shell material should be expressed as "core@shell". In this material alumina is core and polymer is shell. "gamma-Al2O3@ASMA" is correct. "ASMA encapsulated gamma-Al2O3 supports" should be "ASMA encapsulating gamma-Al2O3 supports".

(3) Catalyst characterization section should have more information.

(3-1) TEM: The sample is 4 mm pellet. How was the thin sample for TEM prepared and loaded onto the grid?
(3-2) XRD: Was the sample crushed into powder? Was the sample reduced? If yes, what was the reduction conditions?
(3-3) FT-IR: The title of Fig. 5 is "FT-IR diagram of polymer shell". Was the surface polymer shaved off from the catalyst? Or, was the whole sample crushed into powder? Which was the measurement mode, transmission mode or diffuse reflectance mode? If transmission mode was used, how was the sample disk prepared?
(3-4) XPS: Describe the sample preparation and pre-treatment. Was the reduced sample exposed to air before measurement? How was the binding energy corrected? The standard method using C1s cannot be used for this measurement because the sample contains carbon and C1s signal is overlapped with Ru signal.
If the sample was crushed into poweder, most surface of the sample was crushed Al2O3 core. The discussion on the covering of Al2O3 with polymer is not plausible. The simple ratio of Al2O3 and polymer may explain the decrease of Al signal.

(4) Comment the ratio of polymer and alumina. TG under air atmosphere or organic elemental analysis can determine this ratio.

(5) The crystallite size in XRD (4-digit) is too accurate. I feel that the conversion and selectivity values with 4-digit are also too accurate.

(6) Figure 6a: The peaks at around 285 eV should be labeled as "Ru 3d3/2 + C1s".

(7) Figure 8b: The preparation method (or commercial source) of the other Ru catalysts should be described in experimental section. BET surface area and Ru dispersion (or Ru particle size) of these catalysts are highly desirable.

(8) For the reaction conditions, the word "aqueous" should be included (20 wt.% aqueous glucose).

Author Response

Dear Reviewer:

I am gratitude to receive your reviewing comments and thanks for your sparing time to read my manuscript. Your comments could be the highly encouragement to our subsequent work in the future. In the following part, I am gladly to respond and answer your comments and suggestions.

Comment 1: Title and abstract: the word "glucose" as substrate should be included.

Response 1: Thanks for your impressive suggestion and the title has been changed into “Efficient Sorbitol Producing Process through Glucose Hydrogenation Catalyzed by Ru Supported Amino Poly (Styrene-co-Maleic) Polymer (ASMA) Encapsulating on γ-Al2O3”. Therefore, the word “glucose” as substrate has been included. The same to the part in abstract.

Comment 2: Expression of the catalyst: The core-shell material should be expressed as "core@shell". In this material alumina is core and polymer is shell. "gamma-Al2O3@ASMA" is correct. "ASMA encapsulated gamma-Al2O3 supports" should be "ASMA encapsulating gamma-Al2O3 supports".

Response 2: We have already corrected it in the corresponding sections of the manuscript.

Comment 3-1: TEM: The sample is 4 mm pellet. How was the thin sample for TEM prepared and loaded onto the grid?

Response 3-1: Thanks for your impressive suggestion, since our sample is 4 mm pellet, we have to grind the sample into mesh-like powders and disperse them into the ethanol for loading on the TEM grid. We have rewritten the Catalyst Characterization part in the corresponding section of the manuscript, and more details have been added into it for accuracy.

Comment 3-2: XRD: Was the sample crushed into powder? Was the sample reduced? If yes, what was the reduction conditions?

Response 3-2: Thanks for your impressive suggestion, since our sample is 4 mm pellet, we have to grind the sample into mesh-like powders for the XRD characterization. As we have mentioned that all the samples should be pretreated by reduction with hydrogen environment, and the active sites, Ru species were inert metal element, which could be stable under ambient environment

Comment 3-3: FT-IR: The title of Fig. 5 is "FT-IR diagram of polymer shell". Was the surface polymer shaved off from the catalyst? Or, was the whole sample crushed into powder? Which was the measurement mode, transmission mode or diffuse reflectance mode? If transmission mode was used, how was the sample disk prepared?

Response 3-3: In this work, the FT-IR we applied is to confirm the existence of the amino and hydroxyl groups on the ASMA shell. Therefore, the samples we used in the FT-IR characterization is only the pristine ASMA shell we prepared without encapsulating on the γ-Al2O3 pellet and being impregnated with Ru. Besides, the measurement mode we chose is the transmission mode. The disk preparation details has been added in the corresponding part of the manuscript.

Comment 3-4: XPS: Describe the sample preparation and pre-treatment. Was the reduced sample exposed to air before measurement? How was the binding energy corrected? The standard method using C1s cannot be used for this measurement because the sample contains carbon and C1s signal is overlapped with Ru signal.

If the sample was crushed into powder, most surface of the sample was crushed Al2O3 core. The discussion on the covering of Al2O3 with polymer is not plausible. The simple ratio of Al2O3 and polymer may explain the decrease of Al signal.

Response 3-4: Thanks for your suggestion. I should apologize to our cursoriness of the neglect of some details of characterization when we are writing the part of Materials and Methods in our manuscript. The sample preparation, pre-treatment, binding energy correction method, etc. have been supplemented in our revised version. However, as we have mentioned in the response 3-2, all the samples for characterization have been reduced by hydrogen. Considering for the inert property of Ru, we used it directly in the XPS measurement. However, for excluding the contamination from external environment, the samples were pretreated under an argon flow for 2 h, while they were measured directly without further activation.

As for the decline of the intensity of Al signal, we agree with the comments of the reviewers and have already corrected it in the corresponding sections of the manuscript.

Comment 4: Comment the ratio of polymer and alumina. TG under air atmosphere or organic elemental analysis can determine this ratio.

Response 4: Thanks for your suggestion, in our supplementary materials we have investigated the thermal properties of two samples, including the Ru/γ-Al2O3@ASMA pellet and the sample subjecting γ-Al2O3 pellet through TGA analysis (see Figure S2). Based on the weight loss of these two samples, the proper ratio of ASMA and γ-Al2O3 pellet can be determined to be 1:2.88. We have added this comment in the corresponding sections of the manuscript.

Comment 5: The crystallite size in XRD (4-digit) is too accurate. I feel that the conversion and selectivity values with 4-digit are also too accurate.

Response 5: Thanks for this suggestion. As for the unification and accuracy, in this paper, the crystallite size in XRD acquired from the Scherrer equation, conversion values as well as selectivity values have been corrected to two decimal places rather than 4-digit.

Comment 6: Figure 6a: The peaks at around 285 eV should be labeled as "Ru 3d3/2 + C1s".

Response 6: We have already corrected it in the corresponding sections of the manuscript.

Comment 7: Figure 8b: The preparation method (or commercial source) of the other Ru catalysts should be described in experimental section. BET surface area and Ru dispersion (or Ru particle size) of these catalysts are highly desirable.

Response 7: Thanks for your impressive suggestion. In this work, all these peer supports were provided by the local commercial suppliers. The details and peer catalysts preparation methods were added into the supplementary materials.

Comment 8: For the reaction conditions, the word "aqueous" should be included (20 wt.% aqueous glucose).

Response 8: We have already corrected it in the corresponding sections of the manuscript.
